# Diversity and Typology of Land-Use Explain the Occurrence of Alien Plants in a Protected Area

**DOI:** 10.3390/plants11182358

**Published:** 2022-09-09

**Authors:** Peter Glasnović, Sara Cernich, Jure Peroš, Manja Tišler, Živa Fišer, Boštjan Surina

**Affiliations:** 1Faculty of Mathematics, Natural Sciences and Information Technologies, University of Primorska, Glagoljaška 8, 6000 Koper, Slovenia; 2Nature Park Strunjan—Javni zavod Krajinski Park Strunjan, Strunjan 152, 6320 Portorož, Slovenia; 3Natural History Museum Rijeka, Lorenzov Prolaz 1, 51000 Rijeka, Croatia

**Keywords:** agriculture, biological invasions, conservation management, land cover, plant traits

## Abstract

Plant life history and functional characteristics play an important role in determining the invasive potential of plant species and have implications for management approaches. We studied the distribution of 24 alien plant taxa in a protected area in relation to different land-uses by applying ordination analyses and generalized linear models. Taxa richness is best explained by the presence of built-up areas, followed by residential areas, marshlands, and agricultural lands with semi-natural formations. The diversity of land-use within the grid cell proved to be an important explanatory factor, being the only significant variable explaining the richness of wood perennials and vines. The richness of annual herbs and seed-dispersed taxa is explained by a similar set of variables, with the exception of residential areas. The richness of invasive species is explained only by agricultural land and the diversity of land-use. The richness of taxa with predominant vegetative dispersal is best explained by built-up, marshland, and seminatural areas along with land-use diversity. When we consider only the presence of plant groups within grid cells, the results are similar. The results of similar studies may provide an important tool for defining sustainable practices and overall conservation management in protected areas.

## 1. Introduction

Globalization of biodiversity is one of the consequences of the ongoing human influence on ecological patterns and processes. Alien plant species, especially invasive ones, are considered one of the greatest threats to both biodiversity and human society. In Europe, invasive plants have been shown to have particularly strong negative impacts on ecosystems, surpassing even the harm caused by invasive animals [1]. Although biotic migrations facilitated by human activities are as old as humanity itself, this trend has accelerated greatly in recent centuries due to the rapid development of transportation, industry, and communications. Intrinsic and extrinsic factors influence the direction and patterns of this process. Life form characteristics, stress tolerance, and reproductive strategies of plants favor the spread and establishment of certain plant species in new environments [2,3]. At the same time, human influences in these environments provide the background for their competitiveness with native species. It is widely recognized that human-induced disturbances play a crucial role in the establishment of non-native species [4,5]. Alien floral richness has proved particularly diverse in urban ecosystems, riverine ecosystems, and small-scale rural ecosystems [6]. In agricultural areas, much of the species’ richness is due primarily to alien species, and to an even greater extent than in urban areas [7]. Therefore, with the projected increase of urban areas [8,9], and the changes and prospects in agriculture [10], we can expect the spread and establishment of alien species leading to increased species diversity in the future, together with a harsh decline in the phylogenetic and functional diversity of native communities [11,12,13].

To halt the loss of biodiversity, a number of different conservation tools have been developed. Protected areas aim to preserve biodiversity within specific geographic boundaries, along with the terrestrial or marine landscape and the human actions that have shaped that diversity throughout history, by applying appropriate conservation management. In order to standardize management objectives worldwide, the IUCN proposed to assign protected areas to different categories (Table 1) according to their characteristics and conservation conditions [14]. Today, protected areas are also recognized as important actors in the identification, control, and management of alien species [15]. They represent a refuge from invasions, which could change due to predicted climate changes [16]. In some cases, however, there is also evidence of over-representation of alien species in protected areas compared to other areas, which should be considered particularly alarming [17]. Protected areas are also sensitive to influences emanating from their surroundings [18]. All this is a consequence of the fact that protected areas are increasingly becoming just a fragment within the mosaic a human-modified landscape [15].

Studies have shown that all management categories can be affected by the pressure represented by alien plant species. The highest number of sites with alien species problems has been reported for categories II, IV, and V [19]. While categories II and IV include more or less natural or near-natural areas or area focused on the conservation of particular habitat or species, category V consists of protected landscape where patterns and process were ruled throughout the history by the interaction of people and nature [14].

While inventories of alien species within protected areas can serve as an indicator of the state of the protected area [15], additional knowledge is needed to recognize the impact of alien species on ecosystems and to prioritize management approaches to control and eradicate them from protected areas [20].

In this study, we aim to identify patterns of occurrence of alien species (only neophytes, plant species introduced to Europe after 1492; e.g., [21]) based on different land-use categories in an IUCN Category IV/V protected area at the northern edge of the Mediterranean in the Istrian Peninsula (Slovenia). Specifically, we ask whether (1) land-uses of higher biodiversity conservation value support lower diversity of alien species than other land-uses, whether (2) there are differences in occurrence based on traits, invasion status or pathways of introduction, and finally, and whether (3) this knowledge can lead to appropriate management guidelines for maintaining favorable conservation status of such areas.

## 2. Results

A total of 24 alien taxa were recorded with 992 occurrences altogether (Table 2). The largest number of records (276) was obtained for three species of the genus *Conyza* (*C. sumatrensis* (Retz.) E.Walker, *C. canadensis* (L.) Cronquist, and *C.*
*bonariensis* (L.) Cronquist), followed by *Aster squamatus* (Spreng.) Hieron. and *Robinia pseudoacacia* L. with 175 and 148 occurrences, respectively. Although *C. sumatrensis* is the most widely distributed species in the study area, we decided to consider it only at the genus level due to difficulties in identification at later stages and similarities in life history and ecology.

Of all the recorded taxa, five are classified as invasive at the national level: *Robinia pseudoacacia*, *Helianthus tuberosus* L., *Ambrosia artemisiifolia* L., *Ailanthus altissima* (Mill.) Swingle, and *Erigeron annuus* (L.) Desf, with the first species being the most numerous in the study area (Table 2). The majority of taxa included in the study are considered naturalized. Some of the plants considered problematic, such as *Parthenocissus quinquefolia* (L.) Planch., *Passiflora caerulea* L., *Phyllostachys* sp., *Ligustrum lucidum* W.T.Aiton, *Broussonetia papyrifera* (L.) L’Hér. Ex Vent., and *Cortaderia selloana* (Schult. & Schult.f.) Asch. & Graebn., are also cultivated in gardens, and their occurrence in nature is probably the result of escapes. This trend is supported by the analysis of distance to private homes and, to some extent, distance to tourist facilities (Figure 1).

The unconstrained analysis (DCA, Figure 2 left) generally identifies two groups that cannot be clearly explained by passively projected gradients of land-use. Only the gradient representing the coast clearly explains the occurrence of *Amorpha fruticosa* L. Supplementary variables account for 20.7% of the variation (adjusted explained variation is 10.8%).

The constrained analysis (CCA, Figure 2 right) and forward selection factors (Table 3) showed that public land-use explained most of the variation (4.6%; *p* = 0.014), followed by marshland (4.0%; *p* = 0.006), coast (2.9%; *p* = 0.022) and built-up areas (2.4%; *p* = 0.04). However, when adjusted *p*-values are considered, only the marshland land-use category shows statistical significance.

We found that the richness in the different categories is distributed differently over the study area (Figure 3). Richness in total taxa (Figure 3A) and of taxa with seed dispersal (Figure 3E) partially overlap, while patterns of occurrence diversity in other categories—woody perennials and vines (Figure 3B), annual herbs (Figure 3C), invasive species (Figure 3D), and taxa with vegetative dispersal (Figure 3F) are more scattered.

The land-use values did not show high collinearity and were, therefore, all used for model fitting. Land-use diversity at both accuracy levels showed a high correlation, so it was included in the models individually. Taxa richness is best explained by the presence of built-up areas along with residential areas, marshlands, and agricultural lands with semi-natural formations such as grasslands and hedgerows (Table 4). An important explanatory factor for taxa richness is the diversity of land-use within the grid cell, which is also the only significant variable explaining the richness of woody perennials and vines. Richness of annual herbs and seed dispersed taxa is explained by a similar set of variables as for taxa overall, with the exception of residential areas. The number of invasive species present in the grid cell is significantly explained only by the presence of agricultural land and the diversity of land-use within the grid cell. Richness of taxa with prevailing vegetative dispersal is best explained by built-up, marshland, and seminatural areas together with the diversity of land-uses.

If we consider only the presence of plant groups within the grid cells, the results are similar. The presence of woody perennials and vines is only well explained by the diversity of land-use within the grid cell (Table 5). Presence of annual herbs is best explained by the presence of agricultural land and semi-natural elements such as grasslands and hedgerows. If we consider agricultural land at a more detailed level, the categories that better explain the presence of therophytes are olive groves and field crops (Table 6). The presence of invasive species is also best explained by the diversity of land-use in the grid cell. Since agriculture was found to be one of the most important explanatory variables while fitting richness to land-use (Table 5), we tested the response of invasive species occurrence to the detailed level of land-use. Olive groves showed a significant effect on the occurrence of invasive species. The presence of taxa with seed dispersal is explained with near statistical significance by agricultural and semi-natural stands, while taxa with vegetative dispersal are significantly explained by marshland and land-use diversity.

## 3. Discussion

Habitat heterogeneity and niche partitioning are among the most important factors impacting taxa richness [42,43,44] and there is no particular reason why this should not relate to the diversity of alien plant species as well. Results of this study suggest that the diversity of land-uses, differences in biological traits and pathways of introductions influence the occurrence of alien species within a protected area.

### 3.1. Land-Use and Diversity of Alien Plants

In general, we observed that cells with more diverse land-use support the highest number of aliens in the study area. In the study area, agriculture is the most prevalent land-use within the landscape park (IUCN category V), while (anthropogenic) coastal wetlands and natural areas—consisting of sub-Mediterranean forest fragments and natural coast—dominate in the two nature reserves (IUCN IV category). Agriculture proved to be an important factor influencing the occurrence of alien species. The results show that the diversity of annual taxa and taxa spread mainly by seeds is promoted by agriculture. A particular significance for the occurrence of annual alien taxa was attributed to the cover of field crops and olive groves. To ensure proper production of agricultural crops, farmers must adopt suitable ways to control weeds on agricultural land. The primary role of protected areas is conservation of biodiversity richness and processes, thus agricultural practices should follow rules that ensure favourable conditions for the environment and the conservation of biodiversity. This is particularly important as we have found that agriculture is a major factor explaining the occurrence of invasive species. However, the role of protected areas consists also in achieving socio-cultural objectives, aimed at sustainable development [45]. An example from a neighbouring region (Friuli-Venezia Giulia, Italy) demonstrated that promoting extensive agricultural practices can maintain native plant diversity while reducing the number of exotic species [46]. Preference should be given to approaches that do not involve chemical control. This should be achieved through appropriate crop rotation strategies or sustainable tillage management, that showed to control the persistence of numerous annual plant species considered as weeds on crop fields [22,23,24,35]. Conversely, promoting cover crops instead of regular tillage is shown to improve microbial communities, leading to better agricultural efficiency while maintaining sustainable practices [47].

Coastal wetlands are also subject to problems resulting from the spread of alien species, leading to species loss and a reduction in functional diversity [13]. The impact of alien plant species on wetlands and riparian zones has been widely reported from many regions in the world [6,48,49]. Coastal marshes proved to be particularly significant for the distribution of alien species within the study area. All categories within this land-use are human-made habitats and have been extensively managed over the past several centuries. Nonetheless, these habitats, due to their biodiversity value, have been identified as requiring special protection and their management is a conservation priority. Our results show that land-uses categorized as marshlands are particularly conducive to the occurrence of annual taxa together with taxa characterized by seed and vegetative dispersal. Some of the taxa that occur in these environments are already known to be problematic or in fact represent invasive species (e.g., *Senecio inaequidens* DC., *Lonicera japonica* Thunb., *Helianthus tuberosus*). Others, such as *Aster squamatus*, are still considered at lower risk, as non-invasive species. However, due to their environmental engineering abilities, they may pose a potential threat to native species and habitats in the projected climate changes [50]. This is of particular concern because some of them abundantly occur in coastal wetlands, where the conservation of halophytic vegetation is a primary focus.

Other coastal habitats, like sand dunes and rocky shores, have been shown to be under pressure due to the spread of alien plant species as well [13]. In Strunjan Landscape Park, the rocky shoreline bears two noxious species that occur on sites with the highest visitor impact (*Amorpha fruticosa*, *Ailanthus altissima*). However, our results did not show a major correlation between the diversity of alien species or their occurrence within this land-use category.

As we postulated, the least impacted areas are the more pristine areas dominated by sub-Mediterranean forest stands and flysch cliffs along the natural shoreline. In general, forest stands are considered to be less susceptible to invasion by aliens than other habitats [51]. However, studies show that European woodlands are susceptible to alien plant invasion, especially in combination with human disturbance, fragmentation, alien propagule pression, and soil nutrients level [52]. Singular non-native species can have cascading effects on forest ecosystems [53]. There are also considerable differences when considering the interior of forests compared to their edges [54], and when considering surrounding land-use [55]. In the case of Strunjan Landscape Park, forests are represented by fragments within the matrix of other land-uses, thus management should seriously consider the potential impacts of invasive alien species.

### 3.2. Traits and Pathways of Introduction

Land-use diversity was the only significant variable explaining the diversity of woody, vine, and invasive species in the models. In the previous chapter, we have already emphasised that different land-uses can influence the distribution of plant groups based on different traits. In addition, the spread of alien species is accelerated in fragmented landscapes, especially when major portions of such areas consist of disturbed or man-made habitats [6,56,57]. This study has clearly shown that the distribution of some taxa is related to proximity to human settlements. Built-up areas and residential areas have been shown to be important descriptors of the occurrence of alien taxa in the study area. By using multiple pathways of introduction, species can reach a wider range of suitable habitats, which has been shown to be particularly important for prioritizing conservation efforts in protected areas [58]. General considerations identify two ways that precede successful naturalization—intentional and unintentional introduction [59]. Horticulture is one of the major pathways for non-native plants into new environments [60]. In horticulture, we can consider both types—intentional—when cultivated plants are deliberately introduced into new environments—or, more commonly, unintentional—when introductions are the result of escapes from gardens or parks. By measuring the distance of occurrences from private gardens or parks in tourist facilities, we have found that a number of species that normally grow as ornamentals have most likely spread as escapees from gardens (e.g., *Passiflora caerulea*, *Parthenocissus quinquefolia*, *Ligustrum lucidum*) or have been intentionally introduced (*Phyllostachys* sp.). Due to projected climate changes, the spread of some of these taxa should be carefully monitored because they have been identified as problematic invasive species in warmer areas of the world [29,33]. For some other species (e.g., *Paspalum dilatatum* Poir.), management of public lands (e.g., parks, loans at tourist facilities) plays an important role in their introduction, establishment, and continued spread [37].

### 3.3. Management of Alien Plants in Protected Areas

As the cornerstone of biodiversity conservation, protected areas should be at the forefront of the fight against biological invasions. Protection in itself will not reduce the threat of alien species, therefore, in-situ conservation of biodiversity by detecting and controlling the spread of alien species should be prioritized in protected areas’ management plans. Raising awareness of alien species issues should be in the focus of their conservation efforts. Because of their high standing in society, protected-area managers should take advantage of this to promote activities aimed at improving the implementation of conservation measures by local inhabitants and other users of the services provided by the area, including visitors [61]. This is particularly important because the integrity of the protected areas is highly dependent on the effectiveness of management outside their boundaries [18,62]. Surveillance and monitoring, as well as the development of staff capacity and response networks, should be another important component of the protected areas’ management plans [61]. Early campaigns usually lead to greater success than battles to eradicate infestations after a critical threshold has been reached [63]. However, prioritization should focus on the more vulnerable areas in order to ensure overall conservation, rather than the control itself [15]. Such a management approach can provide long-term and holistic solutions for biodiversity conservation. Managers should work with researchers to determine whether alien species are driving changes in ecosystems or are merely symptoms of the changing environment [20]. Approaches that consider ecological principles of organisms (e.g., dispersal, propagules, ecophysiology, life history) and habitats (e.g., site availability, disturbance) should form the basis of all management guidelines [64]. To achieve this, a robust network linking conservation managers, scientists, and users of protected areas should be established and maintained. In the changing world, protected areas should provide a suitable and well-managed refuge for biodiversity, that is safe from the threats represented by biological invasions.

## 4. Materials and Methods

### 4.1. Study Area

The Strunjan Landscape Park is a protected area situated in the south-western part of Slovenia on the eastern Adriatic coast (Figure 4). It comprises the area of the Strunjan peninsula, a 200 m wide shoreline that extends over a coastal line of 4 km and the entire bay of Strunjan. According to IUCN, the protected area belongs to categories IV and V and is listed as a Specially Protected Area of Mediterranean Importance (SPAMI) since 2019. The park extends over an area of 428.6 ha and includes two nature reserves and a natural monument. Landscape Park Strunjan was established in 1990 and is managed by a public institution since 2008.

The Strunjan Peninsula is characterized by a hilly landscape where agricultural land and farmhouses form a mosaic landscape, which has developed due to favorable natural conditions such as the submediterranean climate and leeward aspect, that allowed humans to settle and develop traditional economic activities in the area. Agriculture shaped the cultural landscape of the Strunjan Peninsula and has maintained a significant impact on the landscape. In the last decades, the coverage of traditional agricultural land decreased, and practices shifted from non-permanent towards permanent plantations (albeit still extensive) that are presently a predominant element in the area. The importance of traditional agriculture is decreasing as it no longer represents the main source of income for the area’s inhabitants, thus making it less appealing for the new inhabitants. The consequences of such changes are reflected in the abandonment and overgrowth of agricultural lands and in the loss of landscape elements and diversity. Despite being one of the most populated protected areas in Slovenia with a population density ranging from 112 to 336 inhabitants per km^2^ [65], only a smaller number of inhabitants engage in traditional agricultural activities.

A second, economically important activity in the area is tourism. The location of the park is very favorable for touristic activities due to good transport infrastructure, proximity to major coastal towns and the attractive seascape. Each year the park is visited by approximately 300,000 visitors, concentrated in the summer season [65]. The 18 accommodation facilities that exist within the park include hotels, apartments, and private rooms. In addition, 7 facilities are located in the near vicinity of the park area. Two parking lots with a total capacity of 600 vehicles are located within the park borders and are mainly used in the summer months by the daily visitors. Higher human pressure during the summer season represents a certain threat to the terrestrial part of the park.

### 4.2. Data Collection

We collected data on the occurrence of all naturalized or supposedly escaped alien plant species within the boundaries of the protected area between 2017 and 2020. Occurrences of individuals or groups of individuals were recorded using a GPS device (Garmin) in WGS84 coordinate systems and subsequently projected to coordinate system D96TM. Taxonomic status was determined based on regional and European floras [66,67,68,69,70,71]. When possible, we considered taxa at the species level. When taxonomic status was difficult to assess, we considered the taxon at the genus level, especially when it appeared that the characteristics and impact of all species within the genus were similar (e.g., *Conyza* spp.). We determined invasive status based on the register of invasive species in the Republic of Slovenia (Ministry of Environment and Spatial Planning of the Republic of Slovenia). Because plant life history and functional traits play an important role in determining the invasive potential of plant species [3], and have implications for management approaches [64], we focused on two general traits: life form and predominant dispersal mode. To further contribute to an appropriate management approach, we reviewed existing control mechanisms for the management of individual taxa.

To understand the role of land-use in the distribution of alien species, we created a layer of features indicating different categories of land-use. This layer was initially created by combining two existing datasets: the *Agricultural holdings and use unit* (GERK) provided by the Ministry of Agriculture, Forestry and Food of the Republic of Slovenia (2020) and the *Land cadastre* provided by the Surveying and Mapping Authority of the Republic of Slovenia (2019). Missing information was supplemented by additional fieldwork in 2020 and interpretation of digital orthophotos at a 1:5000 scale (Surveying and Mapping Authority of the Republic of Slovenia, 2020). Spatial data were obtained and further completed in D96TM national coordinate systems. We compiled two accuracy levels of land-use data (Table 7).

### 4.3. Spatial Procedures

All data were elaborated over a 200 × 200 m grid. Species richness was calculated as the number of alien taxa occurring within a grid cell. We also calculated the richness and binary occurrence of taxa per grid cell considering invasiveness, life form, and prevailing dispersal mode. All spatial procedures were performed using ArcGIS ver.10.7 (ESRI, Redlands, CA, USA). We calculated the distance of taxa occurrence to private homes and tourist facilities using the ‘Near’ tool. The extent of land-use was calculated by summing the areas of each polygon within the grid cell using the ‘Zonal Statistics as Table’ tool. The land-use in each cell was then expressed as a percentage. We also calculated the number of different land-use categories per grid cell. We considered both levels of accuracy in all land-use calculations.

### 4.4. Statistical Analysis

Distances to private homes and tourist facilities were visualized using boxplots from the R package ‘ggplot2’ [72]. Only taxa with 5 or more occurrence data were considered.

Ordination analyses were performed using Canoco v. 5 (Microcomputer Power, Ithaca, New York, USA) [73]. Unconstrained (DCA—discriminant canonical analysis) and constrained (CCA—canonical correspondence analysis) ordination analyses were performed to explain variation in taxa composition in individual grid cells by specific land-use parameters. To determine the length of the gradients, DCA analyses, detrended by segments, were first performed and the models (linear, unimodal) were used accordingly. The statistical significance of the parameters was tested using the Monte Carlo test with 1000 permutations. To estimate the overall affinity of taxa composition, land-use values were passively projected into the ordination plots. The land-use value was estimated as a weighted average of the indicator values of taxa presence. To understand the contribution of land-use to taxa occurrence, we performed step forward analysis by calculating *p* and adjusted *p* values using pseudo-F statistics and significance calculated using the permuted data. Results of all statistical analyses were considered significant if the probability of the null hypothesis was less than 0.05.

To understand the distribution of alien taxa in the studied area, we calculated generalized linear models (GLM) using land-use data as predictors. Collinearity of predictors was assessed by calculating Pearson correlation coefficients. GLMs with Poisson distribution and log linkage were fitted to examine the relationship between predictor variables and taxa richness, while presence–absence data were examined using GLMs with binomial distribution and logit linkage. We fitted GLMs that included all predictors and included only statistically significant predictors from the first model. Models were compared on the basis of AIC value. Correlations and GLMs were calculated using the R package ‘stats’ [74].

## Figures and Tables

**Figure 1 plants-11-02358-f001:**
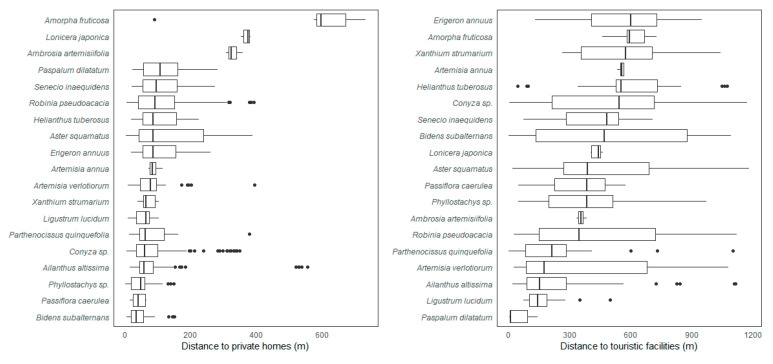
Boxplots indicating minimal, maximal, and median distance (together with outliers) of recorded alien plant taxa to private homes (**left**) and touristic facilities (**right**).

**Figure 2 plants-11-02358-f002:**
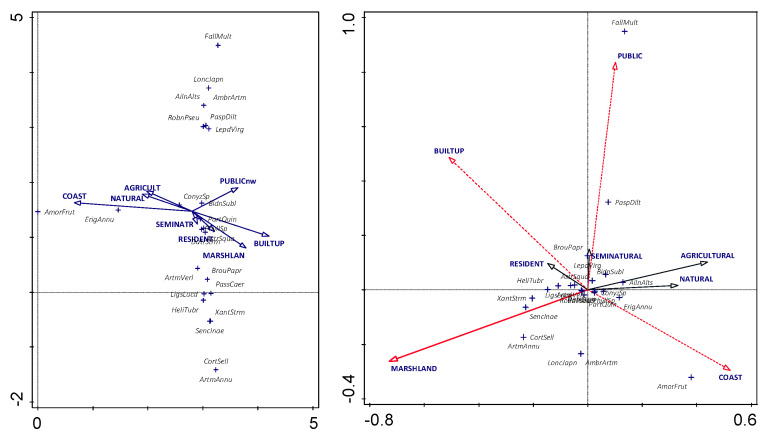
Taxa and factors (land-use), vectors passively projected into the ordination space (**left**). CCA analysis of taxa and factors; red—statistically significant vectors (factors): *p* (adjusted) < 0.05; red-dashed vectors (factors): *p* < 0.05, but *p* (adjusted) > 0.05; black solid vectors (factors)—statistically insignificant vectors (factors): *p* (adjusted) and *p* > 0.05 (**right**).

**Figure 3 plants-11-02358-f003:**
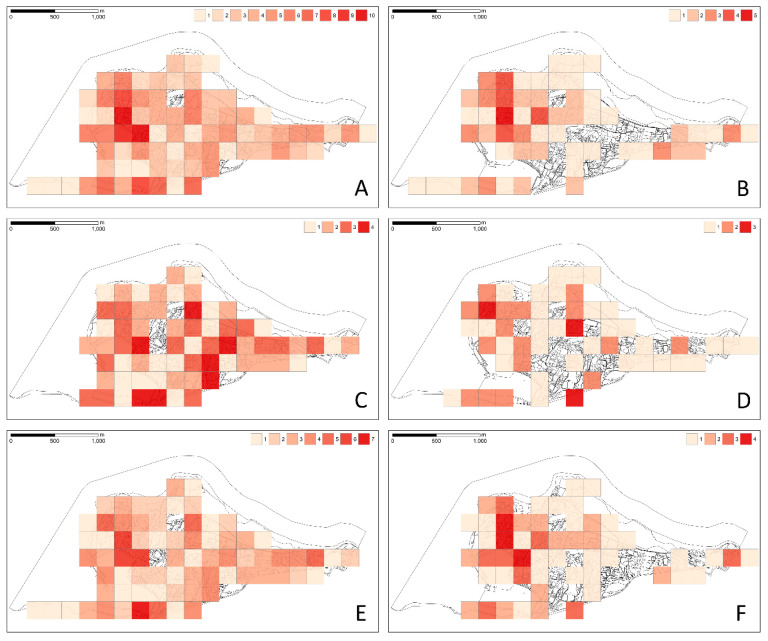
Diversity of aliens in the study area based on different categories expressed as the richness of (**A**)—all taxa; (**B**)—woody perennials and vines; (**C**)—annual herbs; (**D**)—invasive species; (**E**)—taxa with seed dispersal; (**F**)—taxa with vegetative dispersal. Numbers and colors in the legends indicate the number of taxa/species per grid cell.

**Figure 4 plants-11-02358-f004:**
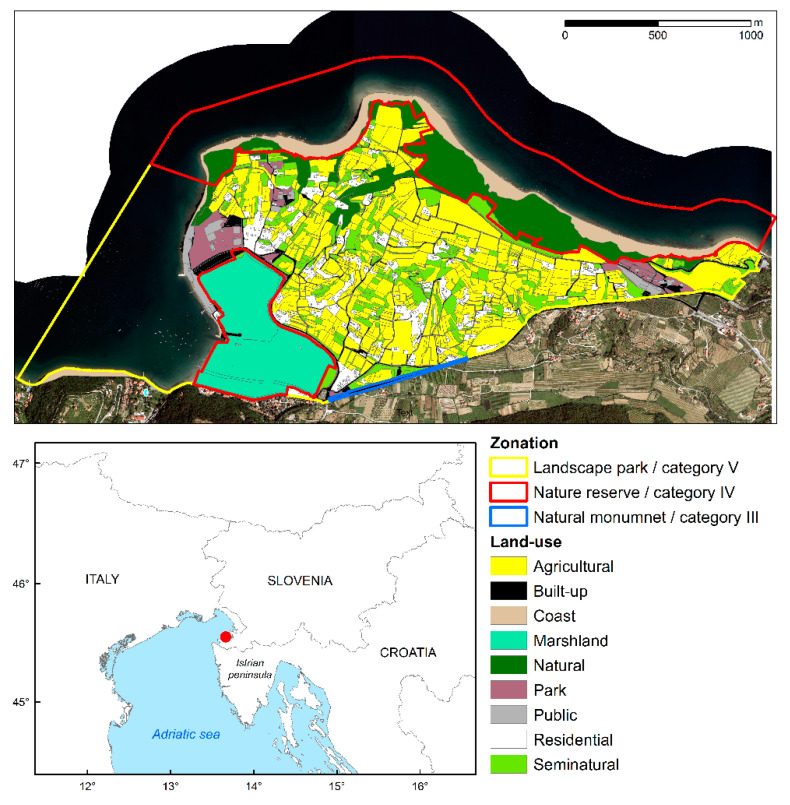
Study area with zonation according to IUCN categories of protected areas and simplified land-use. The red dot indicates the location of the study area within the northern Adriatic region.

**Table 1 plants-11-02358-t001:** IUCN categories of protected areas with definitions [14].

IUCN Category	Definition
I	Strict protection: (Ia) Strict nature reserve and (Ib) Wilderness area
II	Ecosystem conservation and protection (i.e., National park)
III	Conservation of natural features (i.e., Natural monument)
IV	Conservation through active management (i.e., Habitat/species management area)
V	Landscape/seascape conservation and recreation (i.e., Protected landscape/seascape)
VI	Sustainable use of natural resources (i.e., Managed resource protected area)

**Table 2 plants-11-02358-t002:** Alien plant taxa included in the study and basic information about their biology and their management as reported in the literature.

	Taxa	Number of Records	Occurrence in the Protected Area	Status in the Country	Life Form	Prevailing Dispersal Mode	Mechanical Control	General Success
1	*Conyza* sp. (*C. sumatrensis* (Retz.) E.Walker, *C. canadensis* (L.) Cronquist and *C. bonariensis*) (L.) Cronquist	276	naturalized	alien	annual herb	seeds	Requirement of light for germination can address proper management of arable land through mulching or proper tilling strategies. Crop rotation decreases the density of plants. Hand removing of young plants [22,23,24].	Effective
2	*Aster squamatus* (Spreng.) Hieron.	175	naturalized	alien	annual herb	seeds	Hand pulling of plants of all size, preferably before fructification,	No data
3	*Robinia pseudoacacia* L.	148	naturalized/potentially cultivated	alien/invasive	woody perennial	vegetative (root suckers)/seeds	Cuttings, pulling of seedlings [25]	Ineffective due to intensive vegetative resprouting
4	*Artemisia verlotiorum* Lamotte	69	naturalized	alien	perennial herb	vegetative	No data	No data
5	*Ailanthus altissima* (Mill.) Swingle	56	naturalized	alien/invasive	woody perennial	seeds/vegetative	Cuttings, pulling of seedlings [26].	Ineffective due to intensive vegetative resprouting
6	*Bidens subalternans* DC.	46	naturalized	alien	annual herb	seeds	Crop rotation decreases the density of plants. Requirement of light for germination can address proper management through tilling strategies. Frequent cutting (less then 8 weeks) can reduce sexual reproduction capacity [23,27,28].	No data
7	*Phyllostachys* sp.	36	cultivated/escaped/naturalized	alien	perennial herb	vegetative	Intensive and repetitive digging and removing of rhizome and root system [29]	Effective
8	*Parthenocissus quinquefolia* (L.) Planch.	29	cultivated/naturalized	alien	vine	seeds/vegetatively	Cuttings and root digging [30].	Effective
9	*Helianthus tuberosus* L.	25	naturalized/potentially cultivated	alien/invasive	perennial herb with tubers	vegetative (rhizomes)	Regular mowing before new tubers have formed and mowing in riparian habitats [31].	Effective
10	*Senecio inaequidens* DC.	23	naturalized	alien	perennial herb	seeds/vegetative (very rarely)	Hand pulling/mowing ahead seed formation [32].	Effective/moderately effective
11	*Ligustrum lucidum* W.T. Aiton	21	cultivated/escaped	cultivated/alien	woody perennial	seeds/vegetative	Hand removal of young plants. Cuttings of older plants result in vegetative resprouting [33].	Ineffective
12	*Erigeron annuus* (L.) Desf	19	naturalized	alien/invasive	annual herb	seeds	Selective mowing [34].	Moderately effective
13	*Xanthium strumarium* L.	11	naturalized	alien	annual herb	seeds	Seed densities in seed banks greater in conventional tillage than in no tillage areas [35].	No data
14	*Artemisia annua* L.	10	naturalized	alien	annual herb	seeds	No data	No data
15	*Passiflora caerulea* L.	10	cultivated/escaped	cultivated/alien	vine	vegetative/seeds	Hand removal with root system.	No data
16	*Ambrosia artemisiifolia* L.	9	naturalized	alien/invasive	annual herb	seeds	Hand pulling of young plants or very frequent cuttings [36].	Effective/very expensive
17	*Paspalum dilatatum* Poir.	9	naturalized	alien	perennial herb	seeds/vegetative	Intensive mowing reduces lateral spread of rhizomes [37].	Effective
18	*Amorpha fruticosa* L.	7	naturalized	alien	woody perennial	seeds/vegetative	Systematic and repeated cuttings [38].	Effective
19	*Lonicera japonica* Thunb.	5	naturalized	alien	vine	vegetative/seeds	Cuttings results in resprouting of original plant and runners, however cutting and hand removing of spreading colonies may slower the spreading process [39].	Ineffective
20	*Broussonetia papyrifera* (L.) L’Hér. Ex Vent.	3	cultivated/escaped	cultivated/alien	woody perennial	seeds/vegetative	Cuttings, pulling of seedlings [40].	Ineffective due to vegetative resprouting
21	*Lepidium virginicum* L.	2	naturalized	alien	annual herb	seeds	Selective mowing, tends to propagate on barren grounds [41].	No data
22	*Cortaderia selloana* (Schult. & Schult.f.) Asch. & Graebn.	1	cultivated/escaped	alien	perennial herb	seeds	Intensive and repetitive digging of root system before flowering [29].	No data
23	*Datura stramonium* L.	1	naturalized	alien	annual herb	seeds	Hand or mechanical removal of young plants [29].	Effective
24	*Fallopia multiflora* (Thunb.) Haraldson	1	naturalized	alien	vine	vegetative	Cuttings, intensive and repetitive digging and removing of rhizome and root system [41].	No data

**Table 3 plants-11-02358-t003:** Forward selection results of the CCA analysis. Total variation is 4.66, explanatory variables account for 20.7% (adjusted explained variation is 10.8%).

Name	Explains %	Contribution %	Pseudo-F	*p*	*p* (Adjusted)
Public	4.6	22.3	3.4	0.014	0.112
Marshland	4.0	19.4	3.1	0.006	0.048
Coast	2.9	14.1	2.3	0.022	0.176
Built-up	2.4	11.7	1.9	0.04	0.32
Seminatural	2.1	9.9	1.6	0.066	0.528
Agricultural	1.9	9.2	1.5	0.082	0.656
Natural	1.7	8.2	1.4	0.152	1
Residential	1.1	5.2	0.9	0.45	1

**Table 4 plants-11-02358-t004:** Richness (count data) of all taxa, woody perennials and vines, annual herbs, invasive species, and taxa considered based on predominant seed or vegetative dispersal explained by simplified categories of land-use and land-use diversity. Coefficients of generalized linear models (Poisson distribution) selected according to the lowest AIC value. Statistically significant values are indicated with asterisks * *p* ≤ 0.05; ** *p* ≤ 0.01; *** *p* ≤ 0.001.

	All Taxa	Woody Perennials and Vines	Invasive Species	Seed Dispersal	Vegetative Dispersal
(Intercept)	−1.569	−2.302 ***	−2.470 ***	−1.947 ***	−2.663 ***
Coast		−0.023			−0.035
Public		−0.008			−0.010
Built−up	0.051 ***	0.017		0.052 ***	0.056 *
Marshland	0.017 ***	0.006		0.018 ***	0.016 **
Residential	0.013 *	0.008			0.011
Agricultural	0.012 ***	0.001	0.009 *	0.013 ***	0.008
Seminatural	0.025 ***	0.009		0.024 **	0.025 *
Natural		−0.003			0.008
Land-use diversity original					
Land-use diversity simplified	0.271 ***	0.413 ***	0.364 ***	0.283 ***	0.305 **

**Table 5 plants-11-02358-t005:** Presence of woody perennials and vines, annual herbs, invasive species, and taxa considered based on predominant seed or vegetative dispersal explained by simplified categories of land-use and land-use diversity. Coefficients of generalized linear models (Binomial distribution) selected according to the lowest AIC value. Statistically significant values are indicated with asterisks * *p* ≤ 0.05; ** *p* ≤ 0.01; *** *p* ≤ 0.001; and nearly significant with ˚.

	Woody Perennials and Vines	Annual Herbs	Invasive Species	Seed Dispersal	Vegetative Dispersal
(Intercept)		−7.762 **	−4.285 ***	−4.650 ***	−3.437 ***
Coast		0.002	−0.028	0.004	
Public		0.006	−0.013	0.195	
Built−up		0.242	0.076	0.118	
Marshland		14.6	0.012	14.120	0.020 *
Residential		−0.05	−0.024	−0.064	
Agricultural	0.012	0.066 **	0.016	0.045 ˚	
Seminatural		0.158 *	0.03	0.158 ˚	
Natural		0.037	0.01	0.010	0.016
Land-use diversity original					0.378 ***
Land-use diversity simplified	0.767 ***	0.82	0.719 **	0.654	

**Table 6 plants-11-02358-t006:** Presence of annual herbs and alien species explained by agricultural categories of land-use. Coefficients of two generalized linear models (Binomial distribution) fitted to all variables (Model 1) and only statistically significant variables (Model 2). Statistically significant values are indicated with asterisks ** *p* ≤ 0.01; *** *p* ≤ 0.001; and nearly significant with ˚.

	Annual Herbs		Invasive Species	
	Model 1	Model 2	Model 1	Model 2
(Intercept)	−1.260 ***	−1.236 ***	−1.268 ***	−1.254 ***
Crop	0.422 **	0.256 **	0.062 ˚	0.0535 ˚
Orchard	−0.065		−0.028	
Mixed	−0.317 ˚		0.155	
Olive	0.110 **	0.076 **	0.043 **	0.042 ***
Vine	−0.080		−0.018	
AIC	118.39	117.8	157.3	153.65

**Table 7 plants-11-02358-t007:** Two levels of land-use categories used as explanatory factors for the diversity and occurrence of aliens in the study area. Land-uses considered of higher priority for biodiversity conservation are indicated with asterisk (*).

Land-Use—Original	Land-Use—Simplified
Olive grove	Agricultural
Field crop	
Vineyard	
Orchard	
Mixed culture	
Built-up	Built-up
Parking area	
Road—connective	
Road—main	
Road—regional	
Natural coastline with cliffs *	Coast *
Marine lagoon *	Marshland *
Saltworks *	
Riparian vegetation *	
Estuary *	
Submediterranean wood *	Natural *
Park	Park
Recreational area	Public
Touristic facilities	
Other public facilities	
Outbuilding	Residential
Private residence	
Yard	
Sea	Sea
Shrubland and hedgerow	Semi-natural
Meadow	

## Data Availability

Not applicable.

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
