# Peer review of "Diversity and Typology of Land-Use Explain the Occurrence of Alien Plants in a Protected Area"

_plants, 2022, doi:10.3390/plants11182358_

Round 1
Reviewer 1 Report
This is an interesting paper detailing the occurrence of different invasive species with differing functional attributes within different habitat types defined by human use. In doing so this provides data on invasive trait associations with conservation reserves in a Mediterranean ecosystem which would be useful to land managers in this ecosystem. As such this is well worth publishing. I guess the relevance of the research to the international invasive plant community is less clear, though the research certainly supports the notion that greater human activity, and greater diversity in human uses of environments, promotes greater invader richness.
The paper is presented well. There are only a few relatively minor suggestions to improve the paper. These detailed below against line numbers.
Line 63, 75-82. Several times in this section and also in Discussion IUCN categories are mentioned but not defined. For the general reader (including myself) I would like some sort of definition of these categories because I am not familiar with them.
Line 92-97. I support the use of hypotheses raised at the end of the Introduction in structuring the key questions addressed by the paper. However, although these hypotheses are loosely addressed in the Discussion, the clear relationship between the discussion points and the 3 main hypotheses is not made explicit. I think the Discussion would benefit from being structured explicitly around the 3 hypotheses, perhaps even using headings.
Line 101. Can you rewrite this sentence. It doesn't make sense. Perhaps it would be better to say instead 'A total of 24 alien taxa were recorded with 992 occurrences in total (Table 1).'
line 64 and 265. That high impact extensive agricultural practices have lower numbers of exotic species compared to what I would think are lower impact traditional agricultural practices is quite interesting and I would think unexpected. I would expect that with more intensive disturbance, use of herbicides and pesticides, and nutrient addition using fertilizers under industrialized agriculture, there would be greater impacts on the environment and therefore higher weed invasion. Extensive agriculture may use more herbicides and machinery to remove invasive plants than more traditional practices. In this case the greatest impacts might be off-site in adjacent habitat where broad application of herbicides and fertilizer into the surrounding environment lead to elevated resources for invaders. Could this be a reason that more diverse habitat types had greater invasive plants, because these are the areas adjacent to areas of extensive agriculture? Such off-site impacts could be a key reason for invasion in surrounding habitats but not inside the agricultural areas where weed control is routinely done.
Line 301. Just to link this section back to the 3 main hypotheses raised in the Introduction, perhaps start this sentence with 'As we postulated' or something like that linking these findings with the original questions raised f the study?
301-333. As mentioned above, I think the Discussion, particularly in this section, but also possibly at the beginning, needs some broader statements about how the results of this study either support or do not support the three hypotheses raised at the end of the introduction. This would help structure the Discussion around those hypotheses. I feel like without this, the Discussion lacks strong focus on the key questions raised.
Author Response
Line 63, 75-82. Several times in this section and also in Discussion IUCN categories are mentioned but not defined. For the general reader (including myself) I would like some sort of definition of these categories because I am not familiar with them.
Response: We included a short table (Table 1) which summarizes the IUCN categories of protected areas.
Line 92-97. I support the use of hypotheses raised at the end of the Introduction in structuring the key questions addressed by the paper. However, although these hypotheses are loosely addressed in the Discussion, the clear relationship between the discussion points and the 3 main hypotheses is not made explicit. I think the Discussion would benefit from being structured explicitly around the 3 hypotheses, perhaps even using headings.
Response: We have divided the discussion into three parts following the research objectives formulated at the end of the introduction. We have made some minor changes in the discussion text and research objectives to develop the ideas more concisely.
Line 101. Can you rewrite this sentence. It doesn't make sense. Perhaps it would be better to say instead 'A total of 24 alien taxa were recorded with 992 occurrences in total (Table 1).'
Response: We rewrote the sentence to make it more clear.
line 64 and 265. That high impact extensive agricultural practices have lower numbers of exotic species compared to what I would think are lower impact traditional agricultural practices is quite interesting and I would think unexpected. I would expect that with more intensive disturbance, use of herbicides and pesticides, and nutrient addition using fertilizers under industrialized agriculture, there would be greater impacts on the environment and therefore higher weed invasion. Extensive agriculture may use more herbicides and machinery to remove invasive plants than more traditional practices. In this case the greatest impacts might be off-site in adjacent habitat where broad application of herbicides and fertilizer into the surrounding environment lead to elevated resources for invaders. Could this be a reason that more diverse habitat types had greater invasive plants, because these are the areas adjacent to areas of extensive agriculture? Such off-site impacts could be a key reason for invasion in surrounding habitats but not inside the agricultural areas where weed control is routinely done.
Response: We did not compare the impact of intensive and extensive practices on the occurrence of alien plants because there is no intensive agriculture in the area (we additionally emphasized this in the Materials and Methods section (Study area)). However, some practices used in farmland management are from intensive agriculture (use of herbicides and pesticides, use of fertilizers), which should be avoided in protected areas. Our goal was to propose management approaches that aim at sustainable land use while being efficient in combating biological invasions. Examples from neighboring regions have shown that promoting extensive agricultural practices (as opposed to intensive ones) can maintain native plant diversity while reducing the number of exotic species, but we could not test this because there is no such intensive agriculture in the area studied. As an additional comment to the reviewer, we would like to explain that in our case extensive agriculture and traditional practices are almost interchangeable.
Line 301. Just to link this section back to the 3 main hypotheses raised in the Introduction, perhaps start this sentence with 'As we postulated' or something like that linking these findings with the original questions raised f the study?
Response: We agree with the reviewewer.
301-333. As mentioned above, I think the Discussion, particularly in this section, but also possibly at the beginning, needs some broader statements about how the results of this study either support or do not support the three hypotheses raised at the end of the introduction. This would help structure the Discussion around those hypotheses. I feel like without this, the Discussion lacks strong focus on the key questions raised.
Response: The issue was already addressed in one of the previous responses. We divided the discussion into three parts as proposed.
Reviewer 2 Report
These are my main comments on the manuscript (plants-1877005) entitled “Diversity and typology of land use explain the occurrence of alien plants in a protected area”. The manuscript investigates the distribution of 24 alien plant taxa in a protected area in relation to different land uses by applying ordination analyses and generalized linear models. In general, results provide an important tool for defining sustainable practices and overall conservation management in protected areas. Following moderate revisions should be incorporated in the manuscript prior to acceptance.
1. I have concerns about the manuscript sections that I believe need to be addressed in order to improve its clarity.
2. A hypothesis for this research is needed.
3. Other revisions could be checked in PDF attached.

Author Response
- I have concerns about the manuscript sections that I believe need to be addressed in order to improve its clarity.
Response: We have divided the discussion into three parts following the research objectives formulated at the end of the introduction. We have made some minor changes in the discussion text and research objectives to improve its clarity.
- A hypothesis for this research is needed.
Response: We believe that research aims are more appropriate than hypotheses to present and discuss the results of our study. As one part of our study involved also a review of existing management pracices we found this approach even more appropriate.
- Other revisions could be checked in PDF attached.
Response: We have corrected all the other revisions as proposed in the pdf.